# Transcriptome and Metabolome Analysis Reveal the Flavonoid Biosynthesis Mechanism of *Abelmoschus manihot* L. at Different Anthesis Stages

**DOI:** 10.3390/metabo13020216

**Published:** 2023-02-01

**Authors:** Jiaqi Hou, Yuhan Zhou, Liping Ran, Yanzhu Chen, Ting Zhang, Bowei Sun, Yimo Yang, Qianzi Sang, Li Cao

**Affiliations:** Agriculture College, Yanbian University, Yanji 133002, China

**Keywords:** *Abelmoschus manihot* L., flavonoids, antioxidant activity, transcriptome, metabolome

## Abstract

*Abelmoschus manihot* L. (HSK) is a rare and endangered species in the wild that grows on the cliffs of deep mountains. As a natural plant, the chemical composition of HSK is relatively complex, which mainly includes flavonoids, organic acids, polysaccharides, and various trace elements with good effects of clearing away heat, anti-inflammatory, analgesic, and calming nerves, and inhibiting tumor cells. In this experiment, different developmental stages of HSK flowers were used for optimization of the flavonoid extraction and determining method. The antioxidant activities, flavonoid accumulation pattern, and synthesis regulatory network were analyzed using biochemistry, RNA-seq, and UPLC-MS/MS. The total content of flavonoids, vitexin rhamnoside, hyperoside, and rutin in HSK flowers at T3 stage (flower wilting) was significantly higher than in T2 (full flowering) and T1 (bud) stages. Compared with T1 and T2, the antioxidant capacity of the T3 flower alcohol extract was also the strongest, including the total reducing ability, DPPH clearance, OH clearance, O^2−^ clearance, and total antioxidant capacity. A total of 156 flavonoids and 47,179 unigenes were detected by UPLC-MS/MS and RNA-Seq, respectively. The candidate genes and key metabolites involved in flavonoid biosynthesis were identified and the regulatory networks were also analyzed in this study. qRT-PCR test further proved that the gene expression level was consistent with the results of RNA sequence data. The relationship between the gene expression and flavonoid accumulation network provides a theoretical basis for the mining and regulation of functional genes related to the flavonoid biosynthesis and metabolism in *Abelmoschus manihot* L.

## 1. Introduction

*Abelmoschus manihot* L., recorded as “Huangshukui” (HSK) in Chinese, is an annual herb belongs the genus of *Abelmoschus*, which is seed propagation species planted in spring and harvested in autumn [1]. Wild HSK mainly grows in a warm and sunny environment at the crevices of deep mountains and cliffs in the Hebei province of China, with about 2 m plant height, thick roots and stems, lignified base, alternate leaves, and palm-shaped leaves [2]. The flowers of HSK are yellow with purple stamens, the petals are stacked in five pieces, each flower weighs about 56 g, and the flowering period is from July to September each year. Both the main and lateral HSK plants can bear fruit that the mature HSK fruit is similar to cotton boll, and the seeds are slightly smaller than mung bean with brown color [3]. HSK, with strong cold, heat, and salt resistance, can live in 40 °C to −10 °C conditions, while it is afraid of waterlogging and excessively arid places [4]. As a natural plant, HSK can be used as a medicine in its whole plant form, and its chemical composition is complex. Studies have shown that HSK has the highest nutritional value and utilization value among more than 200 okra plants, especially the flowers [5,6], and has the functions of anti-aging, whitening, regulating blood lipids, etc.

The entire HSK plant can be used as medicine. It is known as “Giant Panda in the plant world”, “Life-Saving Grass”, “Plant Sea Cucumber”, “Natural Plant Gold”, and so on. At present, the reported chemical composition of HSK includes flavonoids, unsaturated fatty acids, polysaccharides, saponins, volatile oil, alkaloids, and trace elements [7]. HSK has a very wide range of applications in the market, such as food, medicine, beauty, and other fields. Its flowers, stems, roots, and leaves can be dried in the sun, processed into fine powder, made into HSK additive, or added to various pasta to make cakes and coarse grains. Flowers and rhizomes of HSK can also extract spices and essential oils for industry application. The tender fruit can be fried and eaten directly after cold mixing, and the seeds can be processed into oil [8]. Therefore, different tissues of HSK have different application values.

Previous studies have shown that HSK is a variety with high content of natural flavonoids (5.6% by dry weight) in the plant kingdom, and the flowers have the highest content of flavonoids in all tissues and organs of HSK [9,10], which is the highest content plant material among those commonly used for flavonoid extraction. The total flavonoids of HSK have the functions of anti-diabetes, regulating hypertension, lowering blood lipids, resisting stroke, antitumor, anticoagulation, and antioxidation, and play a protective role in human myocardial ischemia and brain injury to a certain extent [10,11,12]. Quercetin-3-acacia glycoside, hyperoside, quercetin, quercetin-3-glucoside, and myricetin are five mainly rich active flavonoids in HSK plants, and hyperin content is the highest [13]. However, few studies have been conducted on the chemical component biosynthesis of HSK flowers. Therefore, we should continue to study the regulation mechanism of flavonoid biosynthesis and the accumulation pattern of flavonoids during HSK flowering.

In this experiment, total flavonoids were extracted at different flowering stages of HSK; four main flavonoids as well as the all types of flavonoids at each flowering stage were also detected by using LC-MS/MS. Finally, the gene regulatory networks related to flavonoid biosynthesis in HSK flowers were analyzed by transcriptome sequencing. The molecular mechanism of flavonoids biosynthesis during HSK flowering was analyzed and elucidated in the present study. Our results can pave the way to further explore the key genes and functions related to the regulation of flavonoid accumulation.

## 2. Materials and Methods

### 2.1. Plant Materials

*Abelmoschus manihot* L. was planted in Shimen Town (129.02° E, 43.03° N), Antu County, Yanbian Korean Autonomous Prefecture, Jilin Province, on 16 May 2020. The flower samples of HSK were obtained in the bud stage, full flowering stage, and flower wilting stage. Among them, the bud stage is T1 period, full flowering stage is T2 period, and flower wilting stage is T3 period, which were all collected on 16 August, 25 August, and 4 September 2020, respectively. The collected samples were put into a self-sealing bag, then fixed with liquid nitrogen and stored in a refrigerator at −80 °C for later use.

### 2.2. Determination of Total Flavonoid Content

The flower samples of HSK in different flowering periods were dried in an electric air-blast drying oven, the drying temperature and time setting were controlled at 60 °C for 24 h, and then the dried samples were crushed and sieved for later use. Then, 0.1 mg/mL rutin standard solution was used to obtain standard curve (mg/mL) under the 510 nm absorbance value (OD) of a spectrophotometer. The content of total flavonoids was determined by spectrophotometric method followed according to the method of previous research but slightly modified [14,15,16,17]: Accurately absorb 2.0-mL sample solution in turn in two colorimetric tubes as blank control and sample, respectively, add ethanol solution corresponding to concentration to 5.0 mL, add 5% Na_2_NO_2_ solution 0.3 mL, shake well, set still for 6 min, then add 10% Al(NO_3_)_3_ solution 0.3 mL, shake well, static 6 min. Finally, add 1 mol/L NaOH solution 4.0 mL and deionized water 0.4 mL, shake well, let rest 15 min, and determine its absorbance (OD) with spectrophotometer at 510 nm wavelength (DR6000, USA/Colorado). The content of total flavonoids was calculated by standard curve.

Total flavonoid content (mg/g) = CVN/m. In the calculation formula, C is the concentration of the sample solution to be measured (mg/mL); V is the volume of the sample solution to be tested (mL); N is the dilution multiple; m is the sample mass (g).

### 2.3. Vitexin Rhamnoside, Hyperin, Rutin, and Quercetin Content Measurement in HSK

Preparation of standard solution: Accurately weigh the standards of vitexin rhamnoside, hyperin, rutin, and quercetin, respectively, dissolve them in methanol to constant volume, prepare the standard solutions with concentrations of 1.0 mg/mL, 1.0 mg/mL, 0.8 mg/mL, and 0.5 mg/mL, respectively, filter them with 0.22 µm microporous filter membrane, and then store them at 4 °C in the dark. According to the previous research methods and our optimized conditions [9], the crude extract of flavonoids in HSK samples was obtained, centrifuged at 4000 r/min for 6 min, and the supernatant was filtered through 0.22 µm microporous membrane, which was the sample solution to be tested, and stored in 4 °C refrigerators for later use. Conditions for chromatographic separation and analysis: Column Promosil C18 (4.6 mm × 250 mm, 5 m), mobile phase of methanol: 0.2% phosphoric acid (50:50), flow rate of 1.0 mL/min, wavelength of 360 nm, column temperature of 35 °C, single calibration. The accurate peak time of each flavonoid was obtained by single standard injection.

### 2.4. Antioxidant Activity of Total Flavonoids from HSK

The total reducing power was determined by following the method of Zhu et al. [18] with slight modifications: Take 1.0 mL of different concentrations of sample extracts into test tubes, add 2.5 mL of phosphate-buffered saline (PBS, 0.2 mol/L, pH 6.6) and 1.0 mL of 1% K_3_[Fe(CN)_6_] solution, respectively, and mix well. Then, react in a 50 °C water bath for 20 min, take out and rapidly cool with ice water, add 2.5 mL of 10% TCA, mix well, centrifuge at 4000 r/min for 10 min, take 2.5 mL of supernatant, add 2.5 mL of distilled water and 0.5 mL of distilled water 0.1% FeCl_3_ solution, let stand for reaction for 10 min, measure the absorbance value A at a wavelength of 700 nm, and directly measure its reducing power with the A700 value. Each sample solution was measured in parallel 3 times.

For the determination of DPPH free radical scavenging ability, refer to the method of Xiong et al. [19] with slight modification. Accurately weigh 5 mg of DPPH and dissolve it in absolute ethyl alcohol, then put it in a 100 mL brown volumetric flask and keep it away from light. Sample group: weigh 2 mL of sample solution with different concentrations and DPPH solution in the test tube, fully mix them, react at room temperature in the dark 30 min, use absolute ethyl alcohol as blank, and measure its absorbance at 517 nm; control group: 2 mL absolute ethyl alcohol instead of the DPPH solution, blank group: 2 mL absolute ethyl alcohol instead of sample solution, other conditions are the same as above. Each sample solution was measured in parallel 3 times, and the determination of DPPH free radical scavenging ability of sample extracts with different concentrations were calculated according to the following formula:DPPH clearance rate (%) = (1 − (A_sample_ − A_contrast_)/A_blank_)) × 100%(1)

For the determination of hydroxyl radical scavenging ability, we refer to the methods of Wang et al. [15] and Guo et al. [16] and slightly modified them. Sample group: add 1 mL of sample extraction solution with different concentrations, salicylic acid-ethanol solution (9 mmol/L), FeSO_4_ solution (9 mmol/L) and distilled water in turn into the test tube, then add the same volume of H_2_O_2_ (8.8 mmol/L) to start the reaction, heat in a water bath at 37 °C for 15 min after the reaction is complete, adjust the distilled water as a blank to zero, and measure the absorbance at 510 nm. Control group: 1 mL distilled water instead of salicylic acid–ethanol solution; blank group: 1 mL distilled water instead of extractive solution, and other conditions are the same as above. Each sample solution was measured in parallel 3 times, and the determination of OH scavenging ability of sample extracts with different concentrations was calculated according to the following formula:OH clearance rate (%) = (1 − (A_sample_ − A_contrast_)/A_blank_)) × 100%(2)

The method of Liao et al. [20] is referred to and slightly modified for determination. Sample group: sample extracts of different concentrations of 1 mL were successively added to the test tube, mixed with 5 mLTris-HCl buffer (50 mmol/L pH 8.2) and then bathed in water at 25 °C for 20 min. One milliliter pyrogallol solution (3 mmol/L) was preheated in advance and mixed with the above mixture 5 min; then, 1 mL HCl (10 mol/L) was added immediately to stop the reaction, and distilled water was used as a blank to zero, and the absorbance value was determined at 325 nm; control group: 1 mL distilled water was used instead of pyrogallol solution; blank group: 1 mL distilled water was used instead of extract, and other conditions were the same as above. Each sample solution was determined in parallel 3 times, and the O^2−^ scavenging ability of different concentrations of sample extracts was calculated according to the following formula:O^2−^ clearance rate (%) = (1 − (A_sample_ − A_contrast_)/A_blank_)) × 100%(3)

The total antioxidant capacity was determined by the FRAP method [21]. Sample group: take different concentrations of sample solution 2 mL, add 6 mL FRAP working solution, shake well, react 10 min in 37 °C water bath, take out and cool, determine the absorbance value under 593 nm wavelength, calculate the FRAP value of each sample solution according to the standard curve, the greater the FRAP value, the stronger the antioxidant activity; blank group: 70% ethanol instead of sample solution was added to FRAP working solution, and each sample solution was determined in parallel 3 times.
FRAP (mmol/L) = (Ai − 0.1258)/0.7739(4)

### 2.5. Metabolome Detection and Analysis by LC-MS/MS

The collected HSK flower samples at different stages were vacuum freeze-dried for ground into powder (100 mg) dissolving in 1.2 mL 70% methanol for extract. Then, we placed the extract in a 4 °C refrigerator overnight. After 12,000 rpm for 10 min centrifugation, the extract supernatant was filtered with 0.22 μm microporous membrane and stored in the injection bottle for UPLC-MS/MS analysis, which included ultra-performance liquid chromatography (UPLC) and tandem mass spectrometry (MS/MS). The liquid phase condition was followed as shown by Zhou et al. [10].

The quality control (QC) of HSK samples were prepared by mixing the sample extract and used to analyze the repeatability of the sample under the same treatment method. The reproducibility of metabolite extraction and detection, that is, technical reproducibility, can be judged by overlapping display analysis of total ion flow maps (TIC) of different quality control samples. Based on the Metware metabolism self-built database and metabolite information public database, the characteristic ions of each substance were screened using triple quadrupole mass spectrometry in multi-reaction monitoring mode (MRM). According to the retention time (RT) and signal strength (cps) of characteristic ions in the detector, the metabolite detection multi-peak map was used for qualitative and quantitative analysis. The mass spectrometry data were processed by the software Analyst version 1.6.3 (SCIEX, Framingham, MA, USA), and the total ion current map (TIC) and MRM metabolite detection multi-peak map (XIC) of mixed sample quality control QC samples were obtained. The organic mass spectrum file under the sample was opened with MultiaQuant software (Version 2.0, SCIEX, Framingham, MA, USA), and the chromatographic peak was integrated and corrected.

### 2.6. Transcriptome Detection and Analysis in HSK

The total RNA of HSK bud (T1), flowering flower (T2), and withering flowers (T3) used in this experiment was extracted using plant RNA rapid extraction kit. The mortar, centrifugal tube, liquid transfer gun, and other test instruments were strictly sterilized. The degradation degree, purity and integrity of RNA samples were determined for library construction using a Small RNA Sample Pre-Kit (Illumina, San Diego, CA, USA), following the manufacturer’s protocol. The total RNA was set as the starting sample, and the connectors at both ends of the Small RNA were added to synthesize reverse transcription cDNA. Then, the target DNA fragments were separated by PCR amplification and PAGE gel electrophoresis, and the cDNA library was obtained by gel cutting and recovery. The HSK transcriptome was sequenced and analyzed by Illumina Hiseq 2500 (Illumina, San Diego, CA, USA) sequencing platform.

The unigene obtained was compared with the gene sequence in each database using Blast software (National Center for Biotechnology Information, Bethesda, MD, USA) (e-value < 0.00001), and the KEGG lineal homology result of the unigene was obtained using KOBAS version 2.0 (KEGG Orthology Based Annotation System, Beijing, China), and then compared with Pfam database to obtain the final unigene annotation information [22]. The differentially expressed genes’ (DEGs) screening conditions were FDR < 0.01 and fold change (FC) ≥ 2 for Gene Ontology (GO) and Kyoto Encyclopedia of Genes and Genomes (KEGG) analysis [23]. According to the results of molecular function, cellular component, biological process annotation, and unigene pathway enrichment analysis, finding out the relevant DEGs is helpful regarding the gene and expression information as a whole network for easy understanding.

### 2.7. RT-PCR Detection

In order to verify the results of differentially expressed gene analysis in RNA-Seq data, the differential genes related to flavonoid biosynthesis were verified using qRT-PCR. Quantitative analysis was carried out by using an improved plant omnipotent RNA extraction kit (QIAGEN, Hilden, Germany) and analytikjenaqTOWER3/G fluorescence quantitative PCR instrument (Hilden, Germany). The sequence of primers was as follows (Appendix A). The reaction procedure was 95 °C 2 min, 95 °C 5 s, and the annealing temperature was 60 °C 30 s 40 min. Finally, the dissolution curve program was added, and actin was used as the internal reference gene. Each sample was repeated three times, and the relative gene expression of each sample was calculated by 2^−△△Ct^.

## 3. Results and Discussion

### 3.1. Phenotype and Total Flavonoids Measurement in HSK

The stages of bud (T1), full flowering (T2), and flower wilting (T3) of HSK were collected in the present study (Figure 1a). The flower buds of HSK at T1 were small, the petals were not expanded, and the outer corolla was yellowish green. At T2 stage, HSK flowers were large, pale yellow petals with purple base of the inner surface, about 12 cm in diameter. The stamen column was 1.5–2 cm long, anthers subsessile, purplish black stigma, spatulate discoid. In T3, HSK flowers were wilted and the petals are rewrapped, and the outer corolla was yellowish-purple or rose-red. The color of HSK corolla changed obviously in the three stages, and the content of flavonoids in the flowers also changed significantly. Therefore, we then determined the flavonoid content of these three flowering stages.

The content of total flavonoids in HSK at T1, T2, and T3 stages was determined according to the optimized extraction process mentioned above. The results showed that the content of total flavonoid in T1 was 71.74 mg/g, 152.30 mg/g in T2, and 194.76 mg/g in T3, which was significantly higher in T3 than in T1 and T2. The result indicates that the total flavonoid content changed during the growth and development of HSK flowers. The content of total flavonoids was higher in T3 (florescence) (Figure 1b). Therefore, we further carried out targeted detection of several major flavonoids.

### 3.2. Several Main Flavonoids Detection in HSK Flowers

In order to explore the specific content changes of the main active flavonoids in HSK flowers at different flowering stages, this experiment determined the content of vitexin rhamnoside, hyperoside, rutin, and quercetin by using high-performance liquid chromatography. The standard chromatogram map and the peak time of vitexin rhamnoside, hyperin, rutin, and quercetin are shown in Figure 2a,c,e,g and Appendix A, respectively. The relative standard deviation (RSD) of flavonoids is between 0.16% and 0.66%, indicating that the precision of the instrument used is well followed by Wang et al.’s [24] criterion.

The chromatograms and flavonoid content determined by high-performance liquid chromatography are shown in Appendix A and Figure 2, respectively. There were significant differences in the contents of four flavonoids in HSK flowers at different flowering stages. The highest content of vitexin rhamnoside in the HSK flowering stage (T3) was 2.24 mg/g (Figure 2b). The second was 0.77 mg/g in the full flowering stage (T2), and the lowest was 0.28 mg/g in the flowering bract stage (T1). The contents of hyperin and rutin in HSK flowers were gradually and significantly accumulated during the HSK flowers development and maturation, which have the highest content at the falling flowering stage (T3) with 3.18 mg/g and 4.02 mg/g, respectively (Figure 2d,f). Followed by 1.49 mg/g and 2.52 mg/g at the flowering stage (T2), and 0.77 mg/g and 0.94 mg/g at the flowering stage (T1), respectively. However, quercetin gradually deceased during HSK flower development (Figure 2h). The highest content of quercetin was 3.53 mg/g in the T1 flower stage, the second was 2.00 mg·g^−1^ in the full flowering stage (T2), and the lowest in the flowering stage (T3), 1.83 mg/g. These results indicated that there were significant changes in flavonoids during HSK flower development, which affected the medicinal quality or the formation of the nutritional quality of HSK.

### 3.3. Antioxidant Activity Analysis of Total Flavonoids in HSK

Reducing agents (antioxidants) achieve the purpose of scavenging free radicals with the help of their own reduction, where their reducing ability is positively correlated with antioxidant ability, that is, the stronger the reducing ability, the stronger the antioxidant ability [25]. When the concentration is in a certain range, there is a linear relationship between the concentration of total flavonoids and their reducing ability [26]. The antioxidants in the extract can reduce the trivalent iron in potassium ferricyanide to divalent iron, which further reacts with ferric chloride to form Prussian blue, which can obtain maximum absorbance at the 700 nm wavelength [27]. The A700 value of flavonoid extract in three stages of HSK flowers increased with the increase in extract concentration. At the same level, the A700 value of flavonoid extract in T3 (falling flowering stage) was significantly higher than that in T1 (bract stage) and T2 (full flowering stage). The higher the A700 value, the stronger the reducing ability (Figure 3a).

DPPH is a stable free radical [28]. The flavonoid extract of HSK had an effect on the clearance rate of DPPH. Different concentrations of the extract had a significant effect on the clearance rate of DPPH, and the clearance rate increased gradually with the increase in concentration (Figure 3b). At the same extract concentration, the DPPH scavenging rate of flavonoid solution in T3 (falling flower stage) was higher than that in T1 (flower bract stage) and T2 (full flowering stage). At low extract concentration (0.3 mg/mL), the scavenging rate was 76.75% (Figure 3b).

Hydroxyl radicals (OH) can react with many molecules of an organism through a variety of pathways, such as dehydrogenation and electron transfer, leading to cell mutation or direct necrosis [29]. The scavenging rate of hydroxyl radical is an important index for reflecting the antioxidation of substances [30]. Different concentrations of HSK extract had certain scavenging ability on OH produced in the reaction, and the scavenging rate increased with the increase of extract concentration (Figure 3c). At the same extract concentration, the scavenging effect of T3 (falling flower stage) extract on OH was significantly higher than that in the other two periods (T1 and T2), and the highest scavenging rate was 46.35%.

In the process of metabolism, there is superoxide anion radical, namely O^2−^, which can attack fatty acids, proteins and other biological macromolecules, thus damaging the function and structure of cells [31]. Since these free radicals are closely related to biologic pathological changes and aging, they have received much attention in recent years [32]. Therefore, in this study, the effect of HSK extract on O^2−^ scavenging ability were evaluated as well. The result showed that HSK flower had high O^2−^ scavenging ability in three flowering periods, and the scavenging ability increased with the increase in extract concentration (Figure 3d). The total flavonoids in T3 (falling flower stage) had the strongest scavenging ability to O^2−^. When the concentration was 1.5 mg/mL, the O^2−^ scavenging rate was reached at 53.07%, which was significantly different from that in T1 (flowering stage) and T2 (full flowering stage) (Figure 3d).

In the FRAP method, Fe^3+^-TPTZ is reduced to blue Fe^2+^-TPTZ by antioxidants under the condition of low pH, which has a strong absorption at 593 nm [21]. The standard curve was drawn with 0.1–1.6 mmol/L FeSO_4_ solution instead of the sample, and the regression equation was y = 0.7739x + 0.1258 (R^2^ = 0.9994). At the same level (1.5 mg/mL), the FRAP values of T1 (flower bract stage), T2 (full flowering stage), and T3 (falling florescence) were 1.83, 2.31, and 3.70 mmol/L, respectively. The higher the FRAP value, the stronger the total antioxidant capacity, so the order of total antioxidant capacity is T3 > T2 > T1 (Figure 3e).

### 3.4. Metabolome Analysis of Flavonoids in HSK

In order to further analyze the specific accumulation of flavonoids in each developmental period of HSK flowers, in this experiment, metabolome detection was performed by using LC-MS/MS. In this experiment, the total ion flow map (TIC map) detected by QC sample quality spectrum was superposition-analyzed (Appendix A). The results showed that the TIC curves had high overlap, and the retention time and consistency of peak intensity were good, which also indicated that the signal stability was good when the same sample was detected by mass spectrometry at different times. In addition, it also provides an important guarantee for the repeatability and reliability of the metabolome data. Principal component analysis (PCA) of HSK samples (including quality control samples) was performed to understand the overall metabolic differences between T1 vs. T2, T1 vs. T3, and T2 vs. T3 groups, and the degree of variation among each sample. The PCA result shows that T1 vs. T2, T1 vs. T3, and T2 vs. T3 were obviously separated in the first principal component, indicating that there are indeed differential metabolites in the samples of each group (Appendix A).

A total of 156 metabolites were detected by UPLC-MS/MS, including 38 flavonoids, 66 flavonols, 19 anthocyanins, 7 flavanols and tannins, 4 proanthocyanidins, dihydroflavonoids, dihydroflavonols, 3 chalcone and flavonoid glycosides, and 1 isoflavone (Figure 4a). In addition, the variable importance projection (VIP) and difference multiple value (fold change) of OPLS-DA model were used to screen the differentially accumulated metabolites (DAMs). Therefore, in the present study, the unique flavonoids were identified in T1 vs. T2 (1 DAMs), T1 vs. T3 (14 DAMs), and T2 vs. T3 (14 DAMs) comparison groups (Figure 4b). Compared with T1, 15 flavonoid metabolites were screened in T2. Among them, six DAMs were significantly accumulated, and nine DAMs were significantly downregulated in T2 (Figure 4c). Compared with T1, 40 DAMs were significantly different in T3, of which 13 were significantly accumulated and 27 were significantly downregulated (Figure 4d). Compared with T2, a total of 45 flavonoid metabolites was screened in T3, of which 24 were significantly upregulated in T3 and 21 downregulated in T3 (Figure 4e).

### 3.5. Comparison of DAMs in HSK Flowers at Different Developmental Stages

In the comparison between T1 and T2 phases, five representative upregulated metabolites and five downregulated metabolites were identified (Figure 5a). The upregulated metabolites were tangeretin, nobiletin, 3,5,6,7,8,3′,4′-heptamethoxyflavone, myricetin-3-O-(6″-malony) glucoside, and petunidin-3-O-rutinoside. The downregulated metabolites in the T2 period were isorhamnetin-3-O-sophoroside, pelargonidin-3-O-rutinoside, lutcolin-7-O-rutinoside, chrysoeriol-7-O-rutinoside, and luteolin-O-Malonyl-O-Hexoside-O-rhamnoside.

In the comparison between T1 and T3 phases, 12 representative upregulated metabolites and 6 downregulated metabolites were identified in this study (Figure 5b). Upregulated metabolites were protocatechuic acid, naringenin, epicatechin, tamarixetin, laricitrin, phloretin-2′-O-glucoside, mvricetin-3-O-arabinoside, tamarixetin-3-O glucoside, myricetin-3-0-(6″-malony) gluco, pelargonidin-3-O-rutinoside, petunidin-3-O-rutinoside, and quercetin-7-O-(2″-malony) glucosyl-5-O-glucoside. Downregulated metabolites were chrysoeriol-7-O-(6″-malony) glucoside, kaempferol-3-O-sambubioside, kaempferol-3-O-(2″-p-Coumaroyl) galactoside, kaempferol-3-O-(6″-p-coumaroy I) galactoside, luteolin-7-O-rutinoside, and chrysoeriol-7-O-rutinoside.

In the comparison between T2 and T3 phases, six representative upregulated metabolites and four downregulated metabolites were identified (Figure 5c). Upregulated metabolites were tamarixetin, tamarixetin-3-O-glucoside, nepetin-7-O-alloside, isorhamnetin-3-O-(6′′′-malonylglucoside), tamarixetin-3-O-(6″-malonyl) glucose, and luteolin-O-malonyl-O-hexoside-O-rhamnoside. Downregulated metabolites were nobiletin, limocitrin-3-galactoside, chrysoeriol-7-0-(6″-malonyl) glucoside, and tamarixetin-3-O-glucoside-7-O-rhamnoside.

These results showed that the contents of metabolites in the T2 and T3 HSK groups were all higher than those in T1. The contents of these metabolites in T3 were higher than those in T2, which was consistent with the abovementioned determination results of total flavonoid content T3 > T2 > T1. This result indicates that most the flavonoids continuously accumulated significantly in the HSK flowers during their growth and development for quality formation.

### 3.6. Transcriptome Analysis in HSK at Different Flower Stages

In order to understand the molecular mechanism of flavonoid biosynthesis in HSK at different flowering stages, nine libraries (T1-1, T1-2, and T1-3 in bud stage; T2-1, T2-2, and T2-3 in full flowering stage; T3-1, T3-2, and T3-3 in the falling flower stage) were constructed in this experiment. A total of 69.46 Gb was obtained by sequencing, of which the clean data of each sample reached 6.13 Gb, and the percentage of Q30 base was 92.63% or more (Appendix A). The sequencing error rate was ≤0.1%, which proves that the sequencing data was qualified and the quality was up to the standard. Therefore, the following analysis could be continued further.

The gene set was founded by differentially expressed gene (DEG) analysis. According to the relative expression level between each HSK sample, DEGs can be divided into upregulated genes and downregulated genes in different comparison groups (Figure 6a). The results showed that a total of 17,860 DEGs were identified in three flowering stages of HSK. In the T1 vs. T2 comparison group, there were totals of 5054 DEGs (2774 DEGs upregulated and 2280 DEGs downregulated). A total of 8175 DEGs were significantly expressed in the T1 vs. T3 comparison group, which included 4422 upregulated DEGs and 3753 downregulated DEGs. In the T2 vs. T3 comparison group, 2694 DEGs were upregulated and 2259 DEGs downregulated (Figure 6a). In addition, these DEGs have been analyzed by Venn diagram in three comparison groups. There were 552 DEGs, 456 DEGs, and 13 DEGs only present in T1 vs. T2, T1 vs. T3, and T2 vs. T3 comparison groups, respectively (Figure 6b). A total of 3519 DEGs were shared in these three comparison groups. Therefore, these results indicated that the DEGs identified in this study might be involved in flavonoid biosynthesis and flower development in HSK.

### 3.7. KEGG Analysis of Differentially Expressed Genes in HSK

In order to further study the related DEGs that promote the flavonoid accumulation in HSK flowers at different developmental stages, KEGG functional annotation and pathway enrichment analysis were carried out on the selected DEGs. The top 20 pathways with the most reliable enrichment significance (i.e., Q-value) were selected for DEG enrichment analysis (Figure 7). The DEGs in both T1 vs. T2, T1 vs. T3, and T2 vs. T3 comparison groups were significantly enriched in the plant flavonoid biosynthesis pathway and phenylpropane biosynthesis pathway (Figure 7). In addition, the DEGs were also significantly enriched in plant hormone signal transduction, plant–pathogen interaction, and plant MAPK signaling pathways in HSK at different flower development stages (Figure 7). These results indicated that the DEGs among T1, T2, and T3 flower stages were related to flavonoid biosynthesis and flower development in HSK, which provides a theoretical basis for further exploring flavonoid biosynthesis pathways and floral organ development in plants.

### 3.8. RT-PCR Validation

In order to verify the accuracy of RNA-sequence data, nine DEGs involved in flavonoid and anthocyanin biosynthesis were selected, including c147211.graph_c0, c141277.graph_c0, c127212.graph_c0, c141015.graph_c0, c144425.graph_c0, c136934.graph_c0, c78320.graph_c0, c79046.graph_c0, and c131172.graph_c0. The DEG expression level in the three different flower development stages of HSK were analyzed by qRT-PCR (Appendix A). The results showed that the expression levels of these DEGs were consistent with the results of transcriptome data (FPKM value). Therefore, the reliability of the RNA-seq data analyzed in this study was verified.

### 3.9. Combined Analysis of Transcriptome and Metabolomic Profiles

The correlation analysis of DEGs and DAMs in HSK flowers was carried out with a Pearson correlation coefficient greater than 0.8, and the cluster heat map of the correlation coefficient was drawn in T1 vs. T2, T1 vs. T3, and T2 vs. T3 comparison groups (Figure 8a). The result indicated that there are two types of DAMs (flavonoids and tannins). The DEGs and DAMs in each group have an obvious close relationship between them, which indicates that the accumulation of DAMs in different flower development stages were related to the expression changes of DEGs. These DEGs and DAMs were significantly enriched in flavonoid biosynthesis, anthocyanin biosynthesis, and isoflavone biosynthesis pathways.

In the flavonoid biosynthesis pathway (Ko00941), naringenin, epicatechin, fustin, and phlorizin were noted in each of the comparison phases, and all the three metabolites were significantly accumulated (Table 1, Figure 8b). At the same time, 13 differentially expressed enzyme genes were noted, which encode C3′H, CHS, LAR, ANS, F3′5′H, PGT, CCoAOMT3, HCT, DFR, FLS, ANR, CHI, and PGT1. Among them, seven enzyme genes related to metabolite biosynthesis, namely, K05278 (FLS), K05277 (ANS), K13083 (F3′5′H), K01859 (CHI), K00660 (CHS), K22845 (PGT1), and K08695 (ANR). In the anthocyanin biosynthesis pathway (Ko00942), geranin-3-O-rutinoside, cyanidin-3-O-(6″-O-p-coumaroyl) glucoside, cyanidin-3-O-rutinoside, cyanidin-3-O-(2″-O-glucosyl) glucoside, and cyanidin-3,5-di-O-glucoside were noted in the comparison between T1, T2, and T3 phases (Table 2, Figure 8b). At the same time, BZ1 and UGT75C1 enzyme genes were noted to be upregulation-associated in metabolite biosynthesis. In the isoflavone biosynthesis pathway (Ko00943), an upregulated metabolite (naringenin) was noted in the comparison between T1, T2, and T3 comparison phases (Table 3, Figure 8b). Four kinds of differentially expressed enzyme genes HIDH, CYP81E, VR, and IF7MAT were noted as well. Among them, there are four enzymes gene-related to metabolite biosynthesis, namely, K78602 (HIDH), K110364 (CYP81E), K79309 (IF7MAT), and K125179 (IF7MAT).

## 4. Conclusions

In the present study, metabolome, transcriptome, and biochemical analysis were used to analyze the accumulation pattern of flavonoids and the regulatory mechanism of flavonoids biosynthesis (regulatory network) in different flowering stages of HSK. The total flavonoid content and four main flavonoids (vitexin rhamnoside, hyperin, rutin, and quercetin) were determined to have significant differences contents in HSK at different flower development stages. The contents of total flavonoids, vitexin rhamnoside, hyperoside, and rutin were highest in the T3 stage. On the contrary, the content of quercetin was lowest in the T3 stage. The total reactive oxygen species scavenging capacity and total antioxidant capacity of HSK flower alcohol extract in the T3 stage were also higher than in T2 and T1. In addition, a total of 156 flavonoid metabolites and 47,179 unigenes were detected by UPLC-MS/MS and RNA-Seq, respectively. The results showed that DAMs and DEGs were mainly enriched in flavonoid biosynthesis, anthocyanin biosynthesis, and isoflavone biosynthesis pathways. The expression levels of genes related to the flavonoid biosynthesis pathway were different in T1, T2, and T3. The key genes regulated flavonoid biosynthesis were identified and verified by RT-PCR, which included K05278 (FLS), K05277 (ANS), K13083 (F3′5′H), K01859 (CHI), K00660 (CHS), K22845 (PGT1), and K08695 (ANR). These key regulatory genes and intermediate metabolites of flavonoid biosynthesis provide references for further studies of molecular function through genetic engineering and molecular biology techniques.

## Figures and Tables

**Figure 1 metabolites-13-00216-f001:**
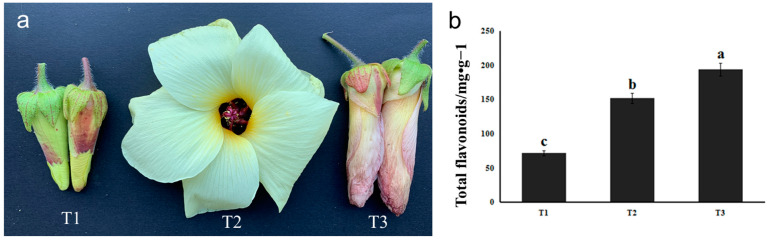
Phenotype (**a**) and total flavonoids measurement (**b**) in HSK at different flower development stages. The different letters (a, b, c) on the columns indicate the significance of the differences between each group, *p* < 0.05.

**Figure 2 metabolites-13-00216-f002:**
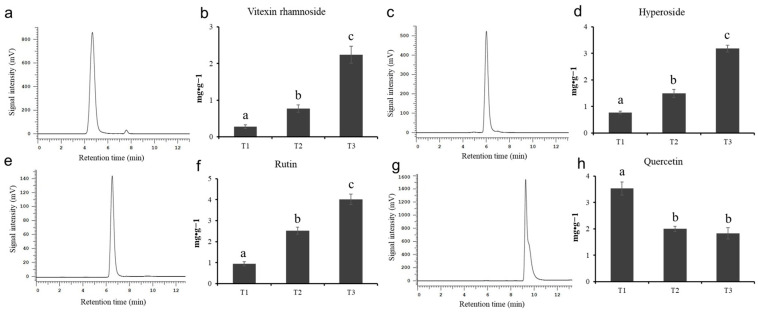
The standard chromatogram map and metabolite content. (**a**,**b**) Vitexin rhamnoside, (**c**,**d**) hyperoside, (**e**,**f**) rutin, (**g**,**h**) quercetin. The different letters (a, b, c) on the columns indicate the significance of the differences between each group, *p* < 0.05.

**Figure 3 metabolites-13-00216-f003:**
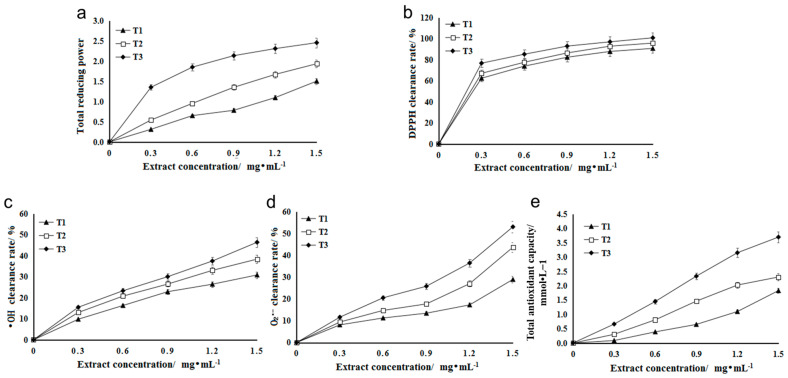
(**a**) The total reducing ability of *Abelmoschus manihot* L. extract. (**b**) Scavenging ability of HSK extract on DPPH. (**c**) Scavenging ability of HSK extract on OH. (**d**) Scavenging ability of HSK extract on O^2−^. (**e**) Antioxidant ability of HSK extract from different flowering periods.

**Figure 4 metabolites-13-00216-f004:**
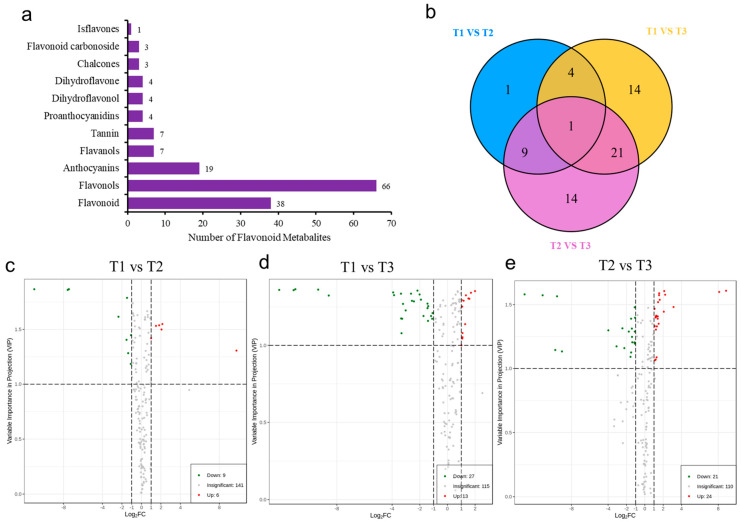
Analysis of flavonoid metabolites in *Abelmoschus manihot* L. (**a**) Types and quantities of metabolites. (**b**) Venn diagram of three HSK comparison groups. (**c**–**e**) Volcano plot of differentially accumulated metabolites (DAMs) in HSK at different flower development stages.

**Figure 5 metabolites-13-00216-f005:**
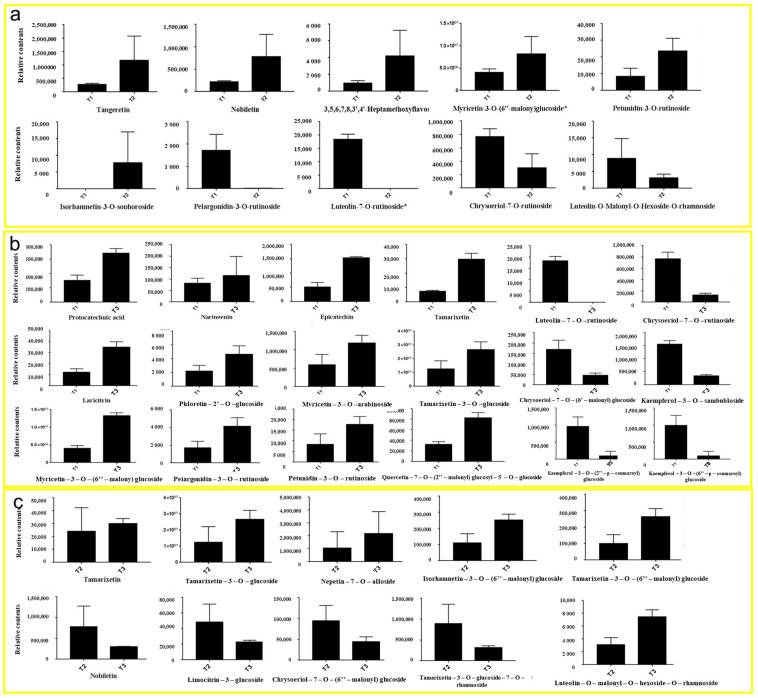
Relative contents of flavonoid and anthocyanin metabolites in HSK at different flowering stages: the bud stage (**a**), full flowering (**b**), and falling flower stage (**c**).

**Figure 6 metabolites-13-00216-f006:**
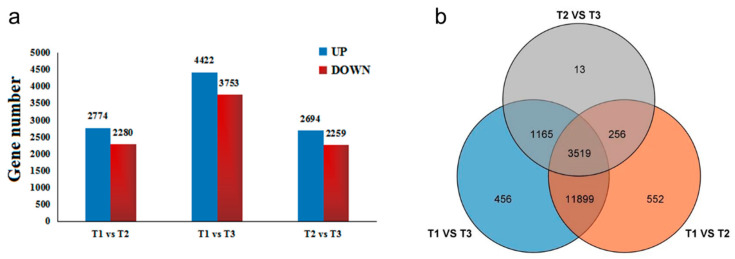
(**a**) Differentially expressed genes (DEGs) among three developmental stages of HSK flowers; (**b**) Venn diagram of differentially expressed genes in three developmental stages of HSK flowers. The UP and DOWN were representing upregulation and downregulation genes, respectively.

**Figure 7 metabolites-13-00216-f007:**
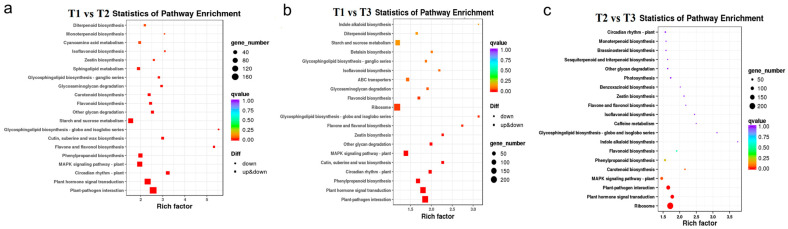
KEGG pathway enrichment of differently expressed genes (DEGs). (**a**) T1 vs. T2 comparison group; (**b**) T1 vs. T3 comparison group; (**c**) T2 vs. T3 comparison group. Diff represents differentially expressed genes. Down and up & down represents downregulated genes and upregulated or downregulated genes, respectively.

**Figure 8 metabolites-13-00216-f008:**
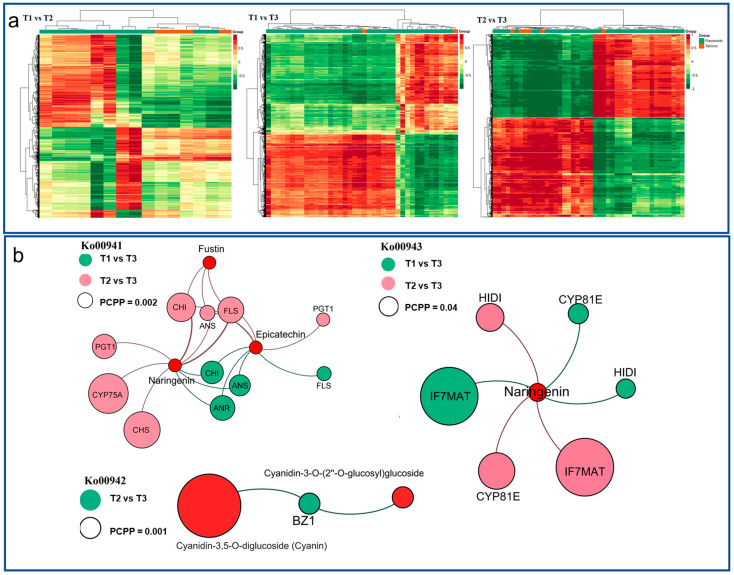
(**a**) Cluster heatmap of correlation between differentially accumulated metabolites (DAMs) and differentially expressed genes (DEGs) at T1 vs. T2, T1 vs. T3, and T2 vs. T3 comparison groups. The abscissa represents the DAMs, the ordinate represents the DEGs. (**b**) Combined transcriptome and metabolome analysis of correlation network patterns in T1 vs. T2, T1 vs. T3, and T2 vs. T3 comparison groups.

**Table 1 metabolites-13-00216-t001:** Differential metabolites and genes in the flavonoid biosynthetic pathway.

Process	Differential Metabolites		Differential Genes			
	Compound ID	Compound	Ko ID	Gene Name	Abbreviation	EC ID
T1 vs. T2	C12644; C08639	Pelargonidin-3-O-rutinoside; Cyanidin-3,5-O-diglucoside (Cyanin)	K12930	anthocyanidin 3-O-glucosyltransferase	BZ1	2.4.1.115
			K12338	anthocyanidin 3-O-glucoside 5-O-glucosyltransferase	UGT75C1	2.4.1.298
T1 vs. T3	C12644; C12095	Pelargonidin-3-O-rutinoside; Cyanidin-3-O-(6″-O-p-Coumaroyl)glucoside	K12930	anthocyanidin 3-O-glucosyltransferase	BZ1	2.4.1.115
			K12338	anthocyanidin 3-O-glucoside 5-O-glucosyltransferase	UGT75C1	2.4.1.298
T2 vs. T3	C12644; C08620; C08639; C16306	Pelargonidin-3-O-rutinoside; Cyanidin-3-O-rutinoside (Keracyanin); Cyanidin-3,5-O-diglucoside (Cyanin); Cyanidin-3-O-(2″-O-glucosyl)glucoside	K12930	anthocyanidin 3-O-glucosyltransferase	BZ1	2.4.1.115

**Table 2 metabolites-13-00216-t002:** Differential metabolites and genes in anthocyanin biosynthesis pathway.

Process	Differential Metabolites		Differential Genes			
	Compound ID	Compound	Ko ID	Gene Name	Abbreviation	EC ID
T1 vs. T3	C00509; C09727; C01604	Naringenin (5,7,4′-Trihydroxyflavanone); Epicatechin; Phloretin-2′-O-glucoside (Phlorizin)	K13065	shikimate O-hydroxycinnamoyltransferase	HCT	2.3.1.133
			K13082	Bifunctional dihydroflavonol 4-reductase	DFR	1.1.1.219
			K05278	flavonol synthase	FLS	1.14.14.82
			K00588	caffeoyl-CoA O-methyltransferase	CCoAOMT	2.1.1.104
			K13081	leucoanthocyanidin reductase	LAR	1.17.1.3
			K05277	anthocyanidin synthase	ANS	1.14.20.4
			K09754	5-O-(4-coumaroyl)-D-quinate 3′-monooxygenase	C3′H	1.14.14.96
			K08695	anthocyanidin reductase	ANR	1.3.1.77
			K01859	chalcone isomerase	CHI	5.5.1.6
			K22845	phlorizin synthase	PGT1	2.4.1.357
			K00660	chalcone synthase	CHS	2.3.1.74
T2 vs. T3	C00509; C01378; C09727	Naringenin (5,7,4′-Trihydroxyflavanone); Fustin; Epicatechin	K13082	Bifunctional dihydroflavonol 4-reductase	DFR	1.1.1.219
			K05278	flavonol synthase	FLS	1.14.14.82
			K13081	leucoanthocyanidin reductase	LAR	1.17.1.3
			K13083	flavonoid 3′,5′-hydroxylase	F3′5′H	1.14.14.81
			K05277	anthocyanidin synthase	ANS	1.14.20.4
			K08695	anthocyanidin reductase	ANR	1.3.1.77
			K01859	chalcone isomerase	CHI	5.5.1.6
			K00660	chalcone synthase	CHS	2.3.1.74
			K13065	shikimate O-hydroxycinnamoyltransferase	HCT	2.3.1.133
			K22845	phlorizin synthase	PGT1	2.4.1.357

**Table 3 metabolites-13-00216-t003:** Differential metabolites and genes in isoflavone biosynthesis pathway.

Process	Differential Metabolites		Differential Genes			
	Compound ID	Compound	Ko ID	Gene Name	Abbreviation	EC ID
T1 vs. T3	C00509	Naringenin (5,7,4′-Trihydroxyflavanone);	K13260	4′-methoxyisoflavone 2′-hydroxylase	CYP81E	1.14.14.89
			K13264	isoflavone 7-O-glucoside-6″-O-malonyltransferase	IF7MAT	2.3.1.115
			K13265	vestitone reductase	VR	1.1.1.348
			K13258	2-hydroxyisoflavanone dehydratase	HIDH	4.2.1.105
T2 vs. T3	C00509	Naringenin (5,7,4′-Trihydroxyflavanone);	K13260	4′-methoxyisoflavone 2′-hydroxylase	CYP81E	1.14.14.89
			K13264	isoflavone 7-O-glucoside-6″-O-malonyltransferase	IF7MAT	2.3.1.115
			K13265	vestitone reductase	VR	1.1.1.348
			K13258	2-hydroxyisoflavanone dehydratase	HIDH	4.2.1.105

## Data Availability

The datasets analyzed during the current study are available in the NCBI BioProject repository, PRJNA786451. The datasets used and/or analyzed during the current study are available from the corresponding author upon reasonable request due to privacy.

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
