# Peer review of "Transcriptome and Metabolome Analysis Reveal the Flavonoid Biosynthesis Mechanism of *Abelmoschus manihot* L. at Different Anthesis Stages"

_metabolites, 2023, doi:10.3390/metabo13020216_

Round 1
Reviewer 1 Report
The authors analyzed the flavonoids content in HSK flowers together with antioxidant capacity. They also identify candidate genes and key metabolites involved in flavonoid biosynthesis. The paper is well written and provide useful information for the scientific community. Therefore I suggest acceptance of the paper in its present form.
Author Response
Reviewer 1:
➢The authors analyzed the flavonoids content in HSK flowers together with antioxidant capacity. They also identify candidate genes and key metabolites involved in flavonoid biosynthesis. The paper is well written and provide useful information for the scientific community. Therefore, I suggest acceptance of the paper in its present form.
Respond: We really thank you for your recognition and support. We will modify and adjust the manuscript according to the journal to meet the publication requirements.
Reviewer 2 Report
Please refer to the attached file.

Author Response
Reviewer 2:
➢Rewrite the sentence lines 14-16 - as it contains four and (In this experiment, different developmental stages of HSK flowers were used 14 for optimization of flavonoids extraction and determining method, and analysis of the antioxidant 15 activities and flavonoids accumulation pattern and synthesis regulatory network by biochemistry, 16 RNA-seq, and UPLC-MS/MS.
Respond: Thank you for your careful reading and review. We have rewritten sentences in line 14-16 according to your comments. Please check again. If there are still any problems, please do not hesitate to contact us, we will try our best to modify it.
➢ In line number 18-19, what are T1, T2 and T3? Please mention
Respond: Thank you for your careful reading and review. Due to our carelessness, we failed to pay attention to these important problems. T1 period is the bud stage, T2 period is full flowering stage, and T3 period is flower wilting stage. We have made a supplementary explanation in the abstract section.
➢ Line number 78 “Abelmoschus manihot L. (HSK)” please remove HSK. Once you mentioned the abbreviation no need to write the full name
Respond: Thank you for your careful reading and review. We have revised the line 78 according to your comments. Please check again.
➢ Please ensure that you have permission to collect your plant. Please add a statement specifying that permissions or licenses were obtained.
Respond: Thank you for your advice and tips. We have added the section "Ethics approval and consent to participate" at the end of the article, which declares the use of plant materials in this manuscript. If there are still any problems, please do not hesitate to contact us, we will try our best to modify it.
➢ Please provide details of the taxonomist who undertook the formal identification of the plant material used in your study.
Respond: Thank you for your suggestion. As we have no previous submission experience, although we have classified and identified the plant materials used in this paper, we have not kept a formal identification report. I (Li Cao) and my colleague Liping Ran participated in the classification and identification of the plant materials used in this experiment and the use permission. In addition, we can provide information on the experts who have formally identified the plant material in this paper: professor Yu Wu, Chengdu Institute of Biology, Chinese Academy of Sciences, e-mail address wuyu@cib.ac.cn; professor Xuhong Song, Chongqing Academy of Chinese Materia Medica, e-mail address songxuhong@cqacmm.com. I am really sorry that we did not pay attention to such problems before. Thank you again for your advice, which let us understand such an important problem. Please forgive us.
➢ How the identification of plant material is done? Information on the voucher specimen must be included in the manuscript.
Respond: Thank you for your suggestion and tips. We have added the section "Ethics approval and consent to participate" at the end of the article, which declares the use of plant materials in this manuscript. Thank you for your professional advice and we hope you can understand us.
➢ Quality of the most figures is low. Please increase the resolution of all figures
Respond: Thank you for your careful reading and review. We have adjusted the resolution of all the figures in the article according to your suggestion and submitted them separately in the system.
➢ Line 91 “Accurately absorb 0- and 2.0-mL sample solution in turn in two…” not clear.
Respond: Thank you for your careful reading and review. You saved us from a major mistake. The 0- should have been removed from the text, but due to our carelessness caused confusion to the reader. We have revised the line 91 according to your comments.

Reviewer 3 Report
Confirmation of the scientific name of the plant is not mentioned?
Vitexin rhamnoside, hyperin, rutin and quercetin content measurement in HS: references?
Replace old sources with new ones.
Using of some other research:
Teucrium polium L. essential oil: Phytochemiacl component and antioxidant properties
International Food Research Journal
|
Chemical composition, antimicrobial and antioxidant properties of essential oil of Origanum vulgar ssp. Gracile |
|
|
Journal of Babol University of Medical Sciences |
Author Response
Reviewer 3:
➢Confirmation of the scientific name of the plant is not mentioned?
Respond: Thank you for your review and suggestions, which are very helpful to improve the quality of our manuscripts. The scientific name of the plant material used in this experiment is Abelmoschus manihot L.
➢Vitexin rhamnoside, hyperin, rutin and quercetin content measurement in HS: references?
Respond: Thank you for your review and suggestions, which are very helpful to improve the quality of our manuscripts. Due to our carelessness, we forgot to add the reference. We have modified the method part of the manuscript according to your suggestion, please check it again.
➢Replace old sources with new ones.
Using of some other research: Teucrium polium L. essential oil: Phytochemiacl component and antioxidant properties. International Food Research Journal.
Chemical composition, antimicrobial and antioxidant properties of essential oil of Origanum vulgar ssp. Gracile. Journal of Babol University of Medical Sciences.
Respond: Thank you for your review and suggestions, which are very helpful to improve the quality of our manuscripts. We have replaced the old literature with “Mahmoudi, R., Nosratpour. S. Teucrium polium L. essential oil: phytochemiacl component and antioxidant properties. Int. Food Res. J. 2013, 20(4):1697-1701.” in the text. However, we could not find the second reference you mentioned. Please forgive us.
